# The Role of Interstitial Brachytherapy for Breast Cancer Treatment: An Overview of Indications, Applications, and Technical Notes

**DOI:** 10.3390/cancers14102564

**Published:** 2022-05-23

**Authors:** Salvatore Cozzi, Matteo Augugliaro, Patrizia Ciammella, Andrea Botti, Valeria Trojani, Masoumeh Najafi, Gladys Blandino, Maria Paola Ruggieri, Lucia Giaccherini, Emanuele Alì, Federico Iori, Angela Sardaro, Sebastiano Finocchi Ghersi, Letizia Deantonio, Cristina Gutierrez Miguelez, Cinzia Iotti, Lilia Bardoscia

**Affiliations:** 1Radiation Oncology Unit, Azienda USL-IRCCS di Reggio Emilia, 42123 Reggio Emilia, Italy; matteo.augugliaro@ausl.re.it (M.A.); patrizia.ciammella@ausl.re.it (P.C.); gladys.blandino@ausl.re.it (G.B.); mariapaola.ruggieri@ausl.re.it (M.P.R.); lucia.giaccherini@ausl.re.it (L.G.); emanuele.ali@ausl.re.it (E.A.); federico.iori@ausl.re.it (F.I.); cinzia.iotti@ausl.re.it (C.I.); 2Medical Physics Unit, Azienda USL-IRCCS di Reggio Emilia, 42123 Reggio Emilia, Italy; andrea.botti@ausl.re.it (A.B.); valeria.trojani@ausl.re.it (V.T.); 3Department of Radiation Oncology, Shohadaye Haft-e-Tir Hospital, Iran University of Medical Science, Teheran 1997667665, Iran; najafi.mas@iums.ac.ir; 4Interdisciplinary Department of Medicine, Section of Radiology and Radiation Oncology, University of Bari “Aldo Moro”, 70124 Bari, Italy; angela.sardaro@uniba.it; 5Radiation Oncolgy Unit, AOU Sant’Andrea, Facoltà di Medicina e Psicologia, Università La Sapienza, 00185 Rome, Italy; sfinocchighersi@ospedalesantandrea.it; 6Radiation Oncology Clinic, Oncology Institute of Southern Switzerland (IOSI), Bellinzona, 6500 Lugano, Switzerland; letizia.deantonio@eoc.ch; 7Brachytherapy Unit, Department of Radiation Oncology, Catalan Institute of Oncology, University of Barcelona, L’Hospitalet de Llobregat, 08908 Barcelona, Spain; cgutierrezm@iconcologia.net; 8Radiation Oncology Unit, S. Luca Hospital, Healthcare Company Tuscany Nord Ovest, 55100 Lucca, Italy; lilia.bardoscia@uslnordovest.toscana.it

**Keywords:** brachytherapy, muticatheter interstitial brachytherapy, APBI, accelerated partial breast irradiation, breast salvage treatment, breast cancer, ipsilateral breast recurrence, brachytherapy boost, breast reirradiation

## Abstract

**Simple Summary:**

Breast cancer is the most common cancer in the female population. Adjuvant radiotherapy has become increasingly important as conservative treatment. Muticatheter interstitial brachytherapy is a type of radiation technique wherein the radioactive sources are directly implanted into or close to the target tissue and may be considered an extremely precise, versatile, and variable radiation technique. Literature data support muticatheter interstitial brachytherapy as the only method with strong scientific evidence to perform partial breast irradiation and reirradiation after previous conservative surgery and external beam radiotherapy. The aim of our work is to provide a comprehensive view of the use of interstitial brachytherapy, with particular focus on the implant description, limits, and advantages of the technique.

**Abstract:**

Breast cancer represents the second leading cause of cancer-related death in the female population, despite continuing advances in treatment options that have significantly accelerated in recent years. Conservative treatments have radically changed the concept of healing, also focusing on the psychological aspect of oncological treatments. In this scenario, radiotherapy plays a key role. Brachytherapy is an extremely versatile radiation technique that can be used in various settings for breast cancer treatment. Although it is invasive, technically complex, and requires a long learning curve, the dosimetric advantages and sparing of organs at risk are unequivocal. Literature data support muticatheter interstitial brachytherapy as the only method with strong scientific evidence to perform partial breast irradiation and reirradiation after previous conservative surgery and external beam radiotherapy, with longer follow-up than new, emerging radiation techniques, whose effectiveness is proven by over 20 years of experience. The aim of our work is to provide a comprehensive view of the use of interstitial brachytherapy to perform breast lumpectomy boost, breast-conserving accelerated partial breast irradiation, and salvage reirradiation for ipsilateral breast recurrence, with particular focus on the implant description, limits, and advantages of the technique.

## 1. Introduction

Breast cancer (BC) is the most common cancer in the female population, accounting for nearly 25% of all cancer diagnoses worldwide, whose incidence has been continuing to grow by approximately 0.5% per year [1]. Despite innumerable advances in medical and radiation oncology, this kind of tumor still represents the second leading cause of death [1].

Adjuvant radiotherapy (RT) has become increasingly important over the years, as conservative treatments for early-stage breast cancer have been demonstrated to offer the same local control of disease and survival outcomes as radical mastectomy compared to surgery alone [2,3,4,5].

Brachytherapy is a type of radiation technique wherein the radioactive sources are directly implanted into or close to the target tissue. In 1922, Geoffrey Keynes first used ‘interstitial radium needles’ for palliative treatment of breast cancer and achieved a surprising ‘disease control in cancer confined to the breast’ with a 3-year survival rate of 83.5% [6]. Nevertheless, breast multi-catheter interstitial brachytherapy (BCT) was systematically introduced in breast oncology practice in the seventies, acquiring an increasingly important role. Currently, breast BCT is the method with the highest scientific evidence and the longest follow-up. Breast BCT may be considered an extremely precise, versatile, and variable radiation technique. Breast BCT has the advantage of delivering high dose levels in the close proximity of the target volume, thus covering the entire tumor bed, and contemporary guaranteeing a very low dose distribution to the organs at risk (skin, heart, and lung), thus providing excellent local control of disease with low toxicity rates, but also requires a high level of expertise [7]. To date, brachytherapy-based accelerated partial breast irradiation (APBI) is the only one with level 1 evidence to be a valid alternative treatment option to whole breast irradiation (WBI) after breast-conserving surgery (BCS) for low-risk, early-stage breast cancer [7,8,9,10]. Moreover, APBI with multi-catheter brachytherapy has also been proposed for adjuvant re-irradiation of in-breast, ipsilateral tumor recurrences after previous BCS and WBI, with a very low rate of side effects and local recurrence rates comparable to salvage mastectomy [11].

The aim of this work is to describe the technical details of the most consolidated breast brachytherapy procedure (that is, multi-catheter, interstitial BCT), the advantages and disadvantages of such a radiation technique, and to evaluate the role of breast BCT in different settings of breast cancer treatment as follows: as lumpectomy boost, APBI, and/or alternative to salvage mastectomy for ipsilateral breast recurrence.

## 2. Implant Technique and Treatment Delivery

### 2.1. Catheter Insertion

The standard procedure for breast catheter insertion consists of a transcutaneous approach. Metallic needles are manually inserted around the open/close cavity created during a lumpectomy, using a plastic guide template with needle holes to achieve geometric dose distribution. The needles are spaced to form equilateral triangles of 12–20 mm, according to the Paris System [12], then inserted in two to four planes, starting from the inferior plane to ensure an acceptable dose coverage to the deep tumor cavity under direct visualization (intraoperative), or guided by ultrasound images (postoperative). The deepest implant plane should be dorsal to the seroma, while the most ventral one should be placed between the skin surface and the seroma. Special care must be taken so that the needles are positioned at a distance of at least 1 cm from the skin surface to avoid late skin toxicity. At the end of the procedure, in the case of an open cavity (seroma), the needles can be replaced by plastic tubes. The number of applicators and tubes varies according to the size of the tumor cavity and breast anatomy (Figure 1) [13,14]. Once the needle positioning has been completed and the adequacy of the implant has been verified, a computed tomography (CT)-based simulation for target volume delineation and radiotherapy planning will be performed. If no appropriate target volume coverage is detected on the simulation CT scan, a few additional catheters may be inserted freehand without the use of a template.

### 2.2. Target Definition and Delineation

Recently, guidelines for patients’ selection and brachytherapy target volume delineation after breast-conserving surgery with both a closed and an open cavity, as well as dose recommendations according to risk factors, were provided by the GEC-ESTRO Breast Cancer Working Group [15,16,17].

A CT scan with a 2–3 mm slice thickness is required to locate the surgical clips, which are needed to properly outline the target volume. Treatment planning begins with the delineation of an estimated target volume, taking into account preoperative imaging (mammography, breast ultrasound, and breast magnetic resonance if available), the surgical scar, the position of the surgical clips, and surgical margins. The clinical target volume (CTV) is defined with the addition of an isotropic, a total safety margin of 20 mm to the estimated target volume, and subtraction of the surgical margin. The thoracic wall and the skin must not be a part of the CTV. No additional margin to obtain the planning target volume (PTV) is necessary if the tumor bed and surgical clips are clearly visible. In the case of uncertainties ranging from 5 to 10 mm, additional margins can be delineated (Figure 2) [18,19].

### 2.3. Dosimetry

The total dose to the target volume is nowadays delivered in the following two different ways: low-intensity pulses repeated every hour for up to a few days (pulse-dose-rate (PDR) brachytherapy); or a few, consecutive, high-dose fractions (HDR), the most used. Various radioisotopes with specific properties in terms of half-life and energy can be used. The most commonly applied in modern brachytherapy are iridium-192, cobalt-60, iodine-125, and palladium-103.

In order to select an appropriate isodose, the dose distribution has to be uniquely normalized. The dwell times are calculated on the basis of volumetric dose constraints. In the case of HDR and PDR BCT, geometric optimization for volume implants should keep the dose non-uniformity ratio (V100/V150) below 0.35 (0.30 ideally) [13]. The volume of PTV receiving 100% of the prescribed dose must be greater than 90% (coverage index ≥ 0.9), with a volume of PTV receiving 150% of the prescribed dose (V150%) less than 30%, and a volume receiving 200% of the prescribed dose (V200%) less than 15%, dose non-homogeneity ratio (V150/V100) < 0.35 (ideally 0.30). The maximum acceptable dose to the skin surface should be less than 70% of the prescribed dose. Table 1 summarizes the GEC-ESTRO normal tissue dose constraints [20] (Table 1).

## 3. Brachytherapy Doses

In 2018, the ESTRO-ACROP expert panel published the following recommendations for breast brachytherapy doses [20].

Recommended radiation schedules for HDR-BCT-based lumpectomy boost are as follows: a biologically equivalent total dose (BED2 for alpha/beta ratio = 4–5 Gy) in the range of 10–20 Gy from 1 to 4 fractions should be selected.

The panel of experts preferably recommends 2 × 4–6 Gy, or 3 × 3–5 Gy scheduled 2 times per day, with an interval between fractions of at least 6 h, and a total treatment time of 1–2 days, or a single fraction of 7–10 Gy, depending on the desired total EQD2.

Recommended schedules for APBI/accelerated partial breast reirradiation (APBrI) with HDR are as follows: 10 fr 3.4 Gy, or 8 fr 4 Gy, or 7 fr 4.3 Gy. With PDR-Brachytherapy: pulsed-dose 0.5–0.8 Gy/pulse, total dose 50 Gy, scheduled every hour, 24 h per day, total treatment time of 4–5 days.

Recommended schedules for lumpectomy boost with PDR-BCT: pulsed-dose 0.5–0.8 Gy/pulse, total dose 10–20 Gy, scheduled every hour, 24 h per day, total treatment time 1–2 days.

## 4. Advantages and Disadvantages of the Technique

The effectiveness of brachytherapy is based on the very high radiation dose directly delivered to the target volume by placing radiation sources in close proximity to or inside the tumor mass/tumor bed. A unique characteristic of this technique is the rapid dose fall-off outside the sources at the end of the implant, thus limiting dose exposure to the surrounding normal tissues. Brachytherapy offers dosimetric advantages with very sharp radiation dose gradients compared to conventional external beam radiation (EBRT) techniques.

As the source moves at the same time as the target, an additional margin is not necessary to cover the set-up uncertainties due to the organ motion, with a subsequent reduction of the planning treatment volume (PTV) and a smaller amount of healthy tissue receiving high doses, hence a reduction in side effects [20]. As a result, brachytherapy combines optimal tumor-to-normal tissue gradients while minimizing the integral dose to the remaining patient’s body tissue [21,22,23,24]. Brachytherapy is preferred in women with large breast sizes and deep tumor masses because the integral dose delivered with electron beams or EBRT is high, with a high risk of unacceptable lung and/or heart dose. Several studies have shown that, from a dosimetric point of view, brachytherapy boost better protects organs at risk (OARs) from medium to high radiation doses in deeply seated lumpectomy beds, compared to EBRT and high-energy electron beams [24]. Actually, brachytherapy is also the radiation technique with the highest level of scientific evidence regarding APBI and APBrI for ipsilateral breast recurrence after curative treatment [25].

Nevertheless, brachytherapy is also burdened with side effects, which may be minor to intense, depending on the delivered dose, the breast tumor site, and the size of the treated volume. Acute reactions (inflammation and irritation at the treatment site) are frequently in view of the very high doses delivered [25,26]. However, the significant decrease in the irradiated volume compared to other radiation techniques contributes to the good long-term functional outcome reported in the literature, with the potential for lower rates of normal tissue fibrosis (which is one of the mechanisms underlying organ dysfunction) [25,26]. Moreover, as it is an invasive treatment, there is a not negligible risk of infection and perioperative pain. The high specialization of the technique, requiring a long learning period to acquire the skills to guarantee the correct positioning of the catheters, may be considered the main limitation of brachytherapy. The Breast BCT procedure also requires specialized equipment able to perform the procedure under aseptic conditions, a dedicated operating room to properly handle the implant, and together with dedicated facilities that meet the radiobiological protection criteria.

## 5. The Roles of Brachytherapy

### 5.1. Brachytherapy as Radiation Boost on the Tumor Bed

Historically, literature data have shown that nearly 40% (70–100%) of breast cancer tumor relapses occur within or close to the tumor bed [2,27,28]. This evidence has given rise to the concept of “dose escalation” on the surgical bed after conservative surgery and WBI to achieve a better local control rate [29,30].

Based on the available results, the GEC-ESTRO Breast Cancer Working Group recommended three categories (low-, intermediate-, and high-risk BC) to drive patients’ selection for lumpectomy boost after WBI [30]. Lumpectomy boost is considered mandatory for high-risk BC patients, including aged ≤40 years, with the presence of close or positive surgical margins, extensive intraductal component, and/or triple-negative phenotype. Lumpectomy boost can be omitted in low-risk BC, including age ≥50 years with unicentric, unifocal, and clear resection margins of at least 2 mm and no axillary lymph node involvement.

The randomized phase 3 trial by Bartelink et al. analyzing 2657 patients with early-stage breast cancer showed a considerable benefit in the boost group in terms of local control rates, particularly in young patients, compared to the no-boost group, but no impact on long-term overall survival (OS) [31]. Moreover, the 20-year cumulative incidence of ipsilateral breast tumor recurrence in the no-boost group was 16.4% vs. 12% in the boost group [31]. Currently, different radiation techniques are available for the administration of the lumpectomy boost contemporary to or after the end of WBI: intraoperative radiotherapy (IORT), photon beam EBRT with conformal, three-dimensional RT (3D-CRT), or intensity-modulated RT (IMRT), or electron beam EBRT with interstitial brachytherapy. Table 2 summarizes the main published series regarding brachytherapy-based radiation boost on tumor beds.

In 1995, Mansfield et al. already demonstrated no difference in local control of disease and progression-free survival between the 654 Ir-192-boosted patients and the 416 electron-boosted patients, nor in the toxicity rates and cosmesis, but there seemed to be an advantage in 10-year local control in patients with stage T1 BC undergoing Ir-192 implant boost [32]. Although not statistically significant, the European Organization for Research and Treatment of Cancer (EORTC) study, where nearly 10% of the treated patients had a brachytherapy boost, showed 2.5% of local recurrences after brachytherapy, compared to 4.7% after an electron boost and 4% after a photon beam EBRT boost. Cosmetic results are excellent/good in 80% of patients [36].

Despite such promising findings, many institutions prefer the use of electron beam or photon beam EBRT boost given its relative ease in set up, reproducibility, outpatient setting, relatively lower costs, reduced execution time for the physician, and excellent results in terms of outcomes and toxicity compared with implants.

In the presence of limited comparative data, brachytherapy-based lumpectomy boost offers the advantage of decreased skin dose and potential radiobiological advantages in terms of outcomes and cosmesis.

Several studies have also provided evidence that brachitherapy boost ensures better OARs sparing from exposure to medium to high radiation doses in deeply seated lumpectomy beds, compared to EBRT and high-energy electron beams.

### 5.2. Brachytherapy to Perform Accelerated Partial Breast Irradiation (ABPI)

In the last thirty years, a broad spectrum of pathological studies has suggested that 80–90% of breast recurrences develop in the initial cancer site [28,37,38]. Although 20% of local recurrences occur within the whole mammary gland, the absolute number of ipsilateral breast recurrences far from the cancer bed is very low, 3–5%, and not influenced by radiotherapy (RT) at all. Indeed, some of them seem to be second-primary tumors [37,39,40]. Recently, the body of evidence from randomized clinical trials supporting the use of APBI after breast-conserving surgery for low-risk early breast cancer (BC) has substantially increased [41]. As a result, APBI is currently an accepted alternative to whole breast irradiation (WBI), in view of comparable survival rates, better cosmesis, and overall treatment time reduction, owing to higher compliance of the treated patients and reduced overall financial treatment costs. The American Society for Radiation Oncology (ASTRO) guidelines recommend APBI after breast-conserving surgery for all cases of invasive ductal/no special type (NST) or ductal in-situ (DCIS) breast carcinoma, with a diameter ≤2 cm (pT1), with negative margins (R0), and negative lymph nodes (pN0), aged ≥50 years, while analyzing case-by-case the opportunity of partial irradiation for 40–49-year-old patients with invasive lobular BC with a diameter from 2.1 to 3 cm [42].

The American Brachytherapy Society (ABS) consensus statement outlined that the strongest evidence in terms of non-inferiority compared to WBI is for multicatheter interstitial brachytherapy- (BCT) and intensity-modulated RT-based APBI [43]. Interstitial BCT remains the APBI technique with the longest follow-up reported. Several studies described adjuvant APBI using the breast BCT technique as not inferior and equally effective as WBI in carefully selected early-stage breast cancer patients (Table 3).

The largest of the phase 3, randomized, equivalence trials is the NSABP B-39/RTOG 0413 study, comparing WBI and APBI delivered with either an external 3D conformal technique, or brachytherapy using a single-entry intracavitary balloon catheter, or interstitial multicatheter brachytherapy [57]. There was an absolute difference in the in-breast tumor recurrence (IBTR) rate between WBI and APBI of 0.4% and 0.5% in the subsets of favorable, early-stage invasive BC and DCIS, respectively, while the absolute difference in the relapse-free interval was 0.1% in the case of low-risk invasive disease and 0.6% for DCIS, respectively [58]. There were no differences in the distant disease-free interval nor in DFS and OS. Cosmetic outcomes and toxicity also resulted in equivalents [59].

The multi-institutional, phase II study RTOG 95-17 Trial reported 5-year actuarial IBTR rates of 4% in the entire cohort, 3% in patients treated with HDR breast BCT, and 6% in the LDR breast BCT groups, good tolerance, and a fairly high rate of good-to-excellent cosmetic results, net of a selection of patients with primary tumors ≤3 cm with clear surgical margins and ≤3 involved axillary nodes with no extracapsular extension [50,51].

The GEC-ESTRO Trial, a European, randomized, phase III multi-institutional study, also met all the non-inferiority criteria and provided comparable outcomes, negligible toxicity, and high-quality cosmetic results of multicatheter brachytherapy than WBI for early-stage invasive BC and in-situ breast carcinoma, with a small absolute difference in local control of 0.5% at 5 years [18,69]. More recently, 5-year results of patient-reported quality of life (QoL) in the same series also supported brachytherapy-based APBI as an alternative treatment option after breast conservative treatment for early-stage BC [70]. Such findings are in line with previous evidence of similar QoL scores after breast-conserving interstitial BCT or WBI in terms of body image, fear of recurrence, and satisfaction with treatment [71,72].

Of note, some of these authors had already identified age <50 years as a predictive factor of local recurrence in a non-randomized, German–Austrian phase II trial since a lower 5-year local recurrence-free survival rate was found in this setting of patients (92.5% vs. 98.9%, *p* = 0.030) [9].

A matched-pair analysis by Wobb and colleagues confirmed brachytherapy-based APBI to have similar IBTR rates, regional recurrence, and contralateral breast failure rates, distant metastasis rates, and similar survival rates to WBI using intensity-modulated RT [52,73]. A comparative study of two prospective cohorts showed dosimetric advantages of multicatheter breast BCT compared to external beam radiation therapy (EBRT), owing to lower doses to the skin, lung, and heart, thus lower acute toxicity, together with a better QoL without compromising tumor control [68].

Cozzi et al. also reported intraoperative interstitial multicatheter HDR BCT to provide high survival rates and a low toxicity profile in both primary and recurrent breast tumor settings, together with good local control and disease control rates [53]. Another, larger series by the same study group seemed to confirm such promising findings [65]. Nevertheless, short courses and single-fraction schedules of breast-conserving BCT-based APBI have been emerging, a harbinger of excellent local control rates, acceptable toxicity, and cosmesis. Although longer-term findings are still scarce, and multi-institutional trials are desirable, the results are consistent and extremely comfortable for patients to better understand future brachytherapy developments [41,74,75].

It should be emphasized that in cases of tumors larger than 3 cm, or the presence of axillary or internal mammary chain node metastases, or in the case of breast implants or expanders, brachytherapy should not be performed, and whole breast external beam radiotherapy is mandatory.

### 5.3. Brachytherapy as Salvage Treatment APBrI for Ipsilateral Breast Recurrence

The association between conservative surgery and adjuvant breast irradiation is actually considered with broad consensus as the standard primary treatment for localized, early-stage breast cancer [3,4,5]. Nevertheless, some patients may experience a second, local, ipsilateral breast tumor event (IBTE), with a reported 20-year cumulative incidence rate of nearly 15% [3]. Salvage mastectomy (SM) has been the treatment of choice for ipsilateral, in-breast tumor relapse (IBTR) for a long time. In fact, several reports of doubling third IBTE rates after further salvage breast-conserving surgery other than salvage mastectomy are available in the literature [76,77,78].

In the era of conservative treatment, a non-demolitive salvage approach, such as a repeated lumpectomy and subsequent re-irradiation for the treatment of IBTR, has been gaining increasing interest, with the added value of not negatively affecting the patient’s QoL and their body perception. Actually, no randomized, phase III comparative studies between salvage mastectomy (SM) and a second-breast conserving treatment (BCS) are available. A recent matched-pair analysis by the GEC-ESTRO Breast Cancer Working Group compared SM (377 patients) and BCS followed by APBrI using interstitial multicatheter brachytherapy (377 patients). With a median follow-up of 75.4 months, no significant differences regarding disease-free survival (DFS), regional and distant relapses, and 5y-OS were reported (88% vs. 87%, respectively, in the last case), as well as third IBTE-free survival rates (2.3% vs. 2.8%, respectively) [79].

Multicatheter breast BCT still represents the most used reirradiation technique in the available literature, whose efficacy has been validated in multiple retrospective series. The main studies on the role of salvage breast BCT as an APBrI are summarized in Table 4. Most of the oldest studies described APBrI with LDR or PDR brachytherapy, while the use of HDR brachytherapy has been homogeneously observed in recent series.

Although most of the reported series are retrospective, with a small number of patients (range from 15 to 217), data on local recurrence (LR) following salvage conservative treatment are encouraging, with third IBTE-free survival rates ranging from 68% to 100% after a median follow up from 19 to 89 months.

The pivotal study in this setting of patients was published by the GEC-ESTRO Breast Cancer Group in 2013 and compared BCT-based APBrI with LDR, PDR, and HDR techniques in combination with salvage lumpectomy [11]. Among the 217 analyzed patients, they reported 5-year and 10-year actuarial second LR rates of 5.6% and 7.2%, respectively; 5-year and 10-year actuarial distant metastases (DM) rates of 9.6% and 19.1%, respectively; 5-year and 10-year actuarial OS rates of 88.7% and 76.4%, respectively. Similar results in terms of freedom from third IBTR greater than 90% were subsequently reported by Smanyko et al., Montagne and colleagues [87,88]. Such findings cleared customed salvage conservative treatment of ipsilateral recurrence of breast cancer.

Regarding safety, mild or moderate toxicity outcomes were usually reported, while ≥Grade 3 side effects ranged from 0% to 16%. Cutaneous and subcutaneous fibrosis, telangiectasia, and hyperpigmentation were predominantly, while ulceration with the unavoidable need for salvage mastectomy was the most frequent severe late adverse event. To support this, the GEC–ESTRO work reported 50% of Grade 1, 39% of Grade 2, 10% of Grade 3, and 1% of Grade 4 (ulceration) late toxicity, respectively [11].

Cosmetic results revealed less clarity to be extrapolated. Satisfactory outcomes were generally reported, with “good” scores in most of the patients, and rare cases described as “poor”. In this regard, the GEC-ESTRO trial reported encouraging results: an “excellent” score for 52 (48%) patients, good for 40 (37%) of them, fair in 14 (13%) cases, and poor for only 2 (2%) patients [11]. However, cosmetic data might be difficult to evaluate in this setting of patients as this is a recurrent disease, which may be assumed to have already been dealt with at least two surgeries and primary, adjuvant external WBI. Despite cosmesis is usually considered satisfactory, patients should be informed of the risk that cosmetic outcomes may be less than optimal with BCT-based APBrI. Nevertheless, patients—and clinicians—may consider this to be an acceptable trade-off given the achieved good survival rates and also considering that the only alternative treatment option is mastectomy (with or without reconstructive surgery).

## 6. Conclusions

Brachytherapy for the treatment of breast cancer is a safe, effective, and extremely versatile radiation technique despite its technical complexity, requiring a high learning curve by the radiation oncologist and dedicated facilities. Since it is an invasive surgical technique and requires considerable costs, breast BCT is performed in a few radiation oncology centers. The main advantage of breast BCT is that it allows complete coverage of the target volume while significantly sparing the surrounding organs at risk. With over 20 years of follow-up, multi-catheter, interstitial brachytherapy represents the BCT technique with the strongest scientific evidence, moreover in different settings of breast cancer treatment. First, brachytherapy has proven to be a good option for lumpectomy boost delivery after WBI, with excellent local control rates and cosmetic results, and low toxicities. A wide range of prospective studies has validated the efficacy of breast BCT in terms of local control of disease and freedom from IBTR also in the setting of accelerated partial breast irradiation after primary breast-conserving surgery, especially for low-risk, early-stage breast cancer, together with tolerable toxicity and good cosmetic results. Short and single courses represent a very comfortable and attractive option. Finally, adjuvant breast reirradiation with BCT may represent a valid alternative to mastectomy as salvage treatment for ipsilateral breast tumor recurrence after primary BCS plus adjuvant WBI: recent data supported brachytherapy-based APBrI following the second lumpectomy as a robust choice, feasible and effective in preventing further local recurrences, with substantially equivalent OS rates to those achieved with salvage mastectomy.

## Figures and Tables

**Figure 1 cancers-14-02564-f001:**
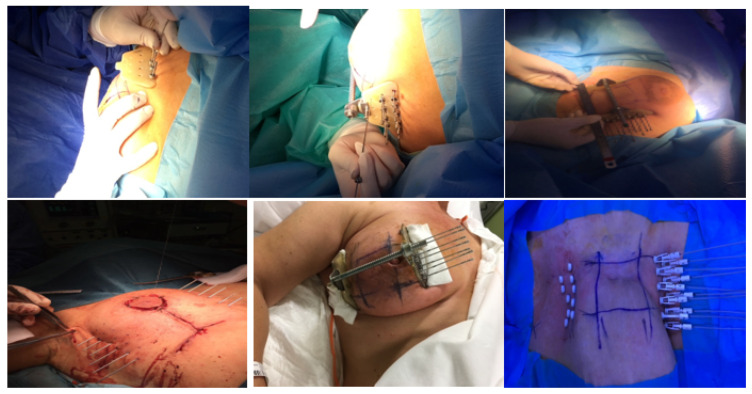
Implant technique: manual insertion of metallic needles.

**Figure 2 cancers-14-02564-f002:**
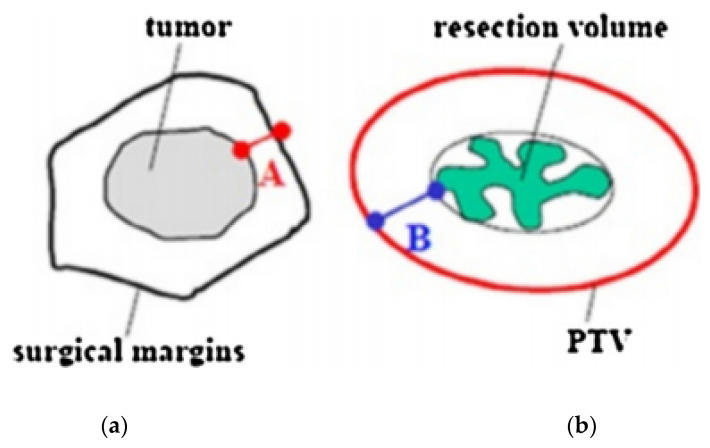
Definition of safety margins. (**a**) Minimal resection margin. (**b**) Safety margin, > 20 mm minus A. PTV: planning target volume.

**Table 1 cancers-14-02564-t001:** Recommended dose–volume limits for OAR-s.

Organs	Constraints
Ipsilateral no target breast tissue	V90 < 10% V50 < 40%
Skin	D1 cm^3^ < 90% D0.2 cm^3^ < 100%
Ribs	D0.1 cm^3^ < 90% D1 cm^3^ < 80%
Heart	MHD < 8% D0.1 cm^3^ < 50%
Ipsilateral lung	MLD < 8% D0.1 cm^3^ < 60%

Abbreviations: MHD: mean heart dose, MLD: mean lung dose. Skin volume is defined as a 5 mm shell below the body contour.

**Table 2 cancers-14-02564-t002:** Main studies concerning brachytherapy lumpectomy boost.

Study	Type of Study	Number of Patients	Follow Up (m)	Total Dose (Gy) (Dose for Fraction) and Technique	Outcomes	Toxicities > G2-3 (%)	Cosmesis Results
Mansfield C.M. et al., 1995 [32].	Retrospective	1070	40	45 + 20 Gy Ir-192 implant	5y-10y LC 93–88%, PFS 93–79%, OS 92–77%	6.5% (18 pts: Moderate/severe fibrosis	91% (575 pts Excellent/good)
Knauerhase 2008 [33].	Retrospective	263	94	electron boost of 6–14 Gy (median, 10 Gy) in 173 patients, with an interstitial boost of 8–12 Gy (median, 10 Gy) in 75 patients (single HDR boost technique), and with a photon (orthovoltage) boost of 7.5–9 Gy in 8 patients.	5- and 10-year LRR of 0% and 5.9%, respectively,	Not reported	Not reported
Polgàr C. et al., 2009 [34].	Prospective randomized study.	207 (52 HDR BT)	63.6	50 Gy WBI + 12–14, 25 Gy HDR BT	5y-LTC (electron boost vs. HDR BT 94.2–91.4%)	7.7 (4 pts: telangiectasia) 17.3 (9 pts: fibrosis)	88.5% (46 pts: Excellent/good,
Poortmans et al., 2004 [35].	prospective randomized multi-center trial.	2661 (225–9% IBT)	60	50 + 15 Gy with an iridium192 implant	5y LF 6 pts 2.5%	7.1% (16 pts: moderate/severe fibrosis	Not reported
Polgar C et al., 2010 [10].	Retrospective	100	94	Single-fraction HDR boost: 8–10.35 Gy (10%) Fractionated multicatheter HDR boost: 3 × 4Gy, 3 × 4.75Gy, 2 × 6.4Gy (90%)	5y LC: 4.5% 8y LC: 7% 8y OS: 80.4%	G3 fibrosis: 6.6% G3 telangiectasia: 2.2%	Excellent good: 56%
Bartelink 2015 [31].	Randomized phase 3 trial	2657	206.4	50 Gy WBI + 15 Gy HDR BT	--	--	--
Quero et al., 2016 [36].	Retrospective	621	122	44 Gy WBI + 5 Gy × 2 fraction HDR BT	10y LR: 7.4% 10y OS: 91% 10y DM 10.6%	Not reported	Excellent good: 80%

Abbreviations: Gy: Gray, pts: patients, y: year; OS: overall survival, 3rdIBTE: third ipsilateral breast tumor event rate, G: grade, MIB: Multicatheter interstitial brachytherapy, LDR: low-dose rate, PDR: pulsed-dose rate, HDR: high-dose rate, y: years, LRR: local recurrence rate; PFS: progression free survival; WBI: whole breast irradiation; LF: local failor; LR: local recurrence; DM: distant metastasis; LTC: local tumor control; LC: local control. Cosmesis results are recorded according to the Harward breast cosmesis scale.

**Table 3 cancers-14-02564-t003:** Summary of the main brachytherapy-based APBI series published in the last twenty years.

Study	Type of Study	Patients	BCT Dose	Outcomes	Toxicity	Cosmesis
Wazer et al., 2002 [44]	Prospective	32	34 Gy (10 fr)	4-y LRR 3%	Skin toxicity: G0-1 30 pts G2 3 pts Subcutaneous toxicity: G0 11 pts G1 3 pts G2 8 pts G 3 3 pts G4 8 pts Fat necrosis 8 pts	0 pts poor 4 pts fair 5 pts good, 24 pts excellent
Perera et al., 2003 [45]	Prospective	39	37.2 Gy (10 fr)	5-y IBTR 16.2%	--	--
Polgar et al., 2004/2013 [7,46]	Prospective	45	30.3 Gy (7 fz) 36.4 Gy (7 fz)	IBTR 6.7% 5-y IBTR rate 4.4% 7-y LRR 9% 7-y CSS 93.3% 7-y RFS 79.8% 10-y LRR 5.9% 10-y OS 80% 10-y CSS 94% 10-y DSS 85%	G1 fat necrosis 20% G2 fat necrosis 2.2% ≥G2 late toxicity 26.7%	84.4% (7y)–81% (10y) excellent/good
Kaufman et al., 2007 [47]	Prospective	32	34 Gy (10 fr)	5-y LRR 6.1% n,3 treatment failures	Fat necrosis 27.3% (2y) 28.1% (5y) 17.9% (>5y) Skin toxicity 28.6% (>5y) G2-3 Subcutaneous toxicity 37.7% (>5y) Pain 17.9% (>5y)	Improved with longer follow up
Wallace et al., 2010 [48]	Prospective	45	28 Gy (4 fr)	--	Acute toxicity: G2 radiation dermatitis 9% G2 breast pain 13% G2 edema 2% G2 hyperpigmentation 2% G3 breast pain 13% Infection 13% Late toxicity: G2 radiation dermatitis 2% G2 induration 2% G2 hypopigmentation 2% G3 breast pain 2% Infection 5% seroma 30% fat necrosis n. 4 rib fractures 4%	96% good/excellent
Strnad et al., 2011 [9]	Prospective	99 HDR BCT 175 PDR BCT	HDR 32 Gy (8 fr) PDR 50 Gy	5-y IBTR 2.9% 5-y LRFS 98% 5-y OS 97% 5-y DFS 96%	≥G3 fibrosis 1(0.4%) ≥G3 telangiectasia 6(2.2%)	90% good to excellent
Shah et al., 2013 [49]	Retrospective	1449	34 Gy (10 fr)	5-y IBTR rate 3.8%	overall fat necrosis rate 2.5% overall infections rate 9.6% overall symptomatic seroma rate 13.4% 2-y symptomatic seroma rate 0.6 %	5-y good/excellent 91.3% 6-y good/excellent 90.5% 7-y good/excellent 90.6 %
Rabinovitch et al., 2014 [50] White et al., 2016 [51] RTOG 95-17	Prospective	65 HDR BCT 33 LDR BCT	HDR 34 Gy (10 fr) LDR 45 Gy (3.5-6d)	10-y IBR 5.2% 10-y LRR 3.1% 10-y DFS 69.8% 10-y OS 78.0%,	G1-2 skin toxicity 78% G3 13% (no G4) skin dimpling/indentation 37% fibrosis 45% telangiectasias 45% skin catheter marks 54% symptomatic fat necrosis 15% Breast asymmetry 73%	66–68% excellent-to-good
Wobb et al., 2016 [52]	Prospective	481 (40% interstitial 60% applicator-based)	Not specified	10-y IBTR rate 4% 10-y DFS 91% 10-y OS 75%	14.4% ≥G2 seroma 12.3% telangiectasia 10.2% symptomatic fat necrosis 5.8% hyperpigmentation 3.3% infection rates	95% good-to-excellent
Strnad et al., 2016 [20] Polgar et al., 2017 [15] GEC-ESTRO Trial	Prospective	633	32 Gy (8 fr) 30.1 Gy (7 fr)	Cumulative incidence of local recurrence 1.44%	No G4 late toxicity 5-y risk of G2-3 skin late toxicity 3.2% 5-y risk of G2-3 subcutaneous tissue late toxicity 7.6% 5-y risk of G3 fibrosis 0% with APBI	93% good-to-excellent
Cozzi et al., 2018 [53]	Retrospective	83 (59 primary BC 24 recurrent BC)	32 Gy (8 fr) 34 Gy (10 fr)	3-y OS 87% recurrent BC 3-y DFS 89% recurrent BC 3-y OS 96% primary BC 3-y DFS 97.8% primary BC No local relapses	Acute toxicity: (1.6%) infectious mastitis (primary BC) 7(30.4%) infectious mastitis and 1(4.3%) hematoma (recurrent BC) Late toxicity: (primary/recurrent BC) G0-2 fibrosis 52(94.4%)/7(31.9%) G3 fibrosis 3(5.6%)/11(50%) mastitis 3(5.5%)/6(27.3%) hypochromic skin spots 8(14.8%)/8(36.4%) skin hyperpigmentation 4(7.4%)/3(13.6%) telangiectasia 1(1.9%)/7(3z1.8%)	Primary BC: 11.1% excellent 63% good 1.8% fair 0% poor 4.9%, no ratings available. Recurrent BC: 63% good 27.3% fair 32% poor 7%, no cosmetic ratings available.
Hepel et al., 2018 [54]	Prospective	40	28.5 Gy (5 fr)	No local relapses	Acute G0-1 skin reaction 70% Acute G2 skin reaction 28% Acute G3 skin reaction 3% No ≥G3 late toxicity	--
Pohanková et al., 2018 [55]	Retrospective	125	34 Gy (10 fr)	No relapses	2(1.8%) wound dehiscence 7(6.2%) inflammatory complications 6(4.4%) G1 radiodermatitis 3(2.7%) seroma No ≥G3 late toxicity	92% excellent or good
Khan et al., 2018 [56]	Prospective	200	22.5 Gy (3 fz)	n.1 IBTR n.1 regional nodal failure	Radiation dermatitis 31(15.5%, G3 1(0.5)) Breast pain 31(15.5%) Breast infection 3(1.5%) Breast edema 2(1%) Superficial tissue fibrosis 12(6%) Deep tissue fibrosis 22(11%) Seroma formation 8(4%) Hyperpigmentation 3(1.5%) Fat necrosis 1(0.5%) Nonhealing wound 2(1%, G3 2(1%)) Fatigue 1(0.5%)	97.25% excellent or good
Vicini et al., 2018 [57,58,59] RTOG 0413	Prospective	2107	34 Gy (10 fr) BCT 38.5 Gy (10 fr) EBRT	90(4%) IBTR 4.6% 10-y cumulative incidence of IBTR	845(40%) G1 toxicity 921(44%) G2 toxicity 201(10%) G3 toxicity	Equivalent between APBI and WBI
Gaudet et al., 2019 [60]	Retrospective	364	32 Gy (8 fr)	n.14 IBTR 5-y OS 95.1% 10-y OS 92.2% 5-y LRFS 96.2% 10-y LRFS 88.8%	--	--
Maranzano et al., 2019 [61]	Prospective	133	32 Gy (8 fr)	3(2%) IBTR 5-y OS 95% 5-y CSS 100% 10-y OS 84.55% 10-y CSS 100% 13-y OS 81.4% 13-y CSS 100%	Late toxicity related to the skin administered dose (≤55% of the PD vs. 55%)	80% excellent
Hannoun-Lévi et al., 2018/2020 [62,63]	Prospective	26	16 Gy (1 fr)	5-y LRFS 100% 5-y MFS 95.5%, 5-y CSS 100% 5-y OS 88.5%	Acute toxicity G1 75.7% G2 22.8% G3 4.5% 31.8% breast fibrosis 13.6% puncture site inflammation 11.4% skin hyperpigmentation Late toxicity 1 G2 cytosteatonecrosis 2 G1 hypopigmentation (puncture site) 1 G1 breast fibrosis 1 G2 breast fibrosis	81% excellent 19% good
Rodriguez-Ibarria et al., 2020 [64]	Prospective	182	32 Gy (8 fr)	5-y LR 1.1% 5-y DFS 97.2% 5-y OS 93.2%	n.1 G2 radiodermitis n.1 G2 hyperpigmentation n.3 G2 acute induration no G3 toxicity no G3-4 late toxicity 9(5.5%) breast induration 1(0.6%) chronic hyperpigmentation 4(2.4%) telangiectasia	--
Laplana et al., 2021 [65]	Retrospective	289	32 Gy (8 fr) 34 Gy (10 fr) 16 Gy (1 fr)	5-y LC 98.9% 5-y DFS 96.7%, 5.y CSS 99.1% 5-y OS 95.6%	14.8% fibrosis 8.8% skin discoloration at the catheter points 0.5% telangiectasia	88.3% excellent or good
Hepel et al., 2021 [66]	Retrospective	252	34–36 Gy (10 fr) 28.5 Gy (5 fr)	2-y LRFS 98.3% 5-y LRFS 90.9%	Acute G0-1 radiodermatitis 77% Acute G2 radiodermatitis 19% Acute G3 radiodermatitis 4% G2 late toxicity 8.8% G3 late toxicity 1%	62% excellent 36% good 2% fair/poor
Polgar et al., 2021 [67]	Prospective	88	36.4 Gy (7 fz)	5-y IBF 4% 10-y IBF 5.8% 20-y IBF 9.6% 5-y DFS 88.8% 10-y DFS 86.2% 20-y DFS 79.7% 5-y OS 93.7% 10-y OS 77.2% 20-y OS 59.5% 5-y CSS 98.4% 10-y CSS 94.9% 20-y CSS 92.6%	G2-3 late skin toxicities 17(13.6%) G2-3 fibrosis 18(14.4%)	20-y 82.4% excellent or good
Garduño-Sánchez et al., 2022 [68]	Prospective	76	32 Gy (4 fz)	Estimated 5-y OS 96.8% 10-y OS 77.7%, 5-y DFS 91.1% 10-y DFS 69.4%	Acute G1-2 dermatitis 51.4% Late thickening of skin 93.3% Late asymmetry 33.3% Late fibrosis 88.9 Late architectural distortion 83.3% Late retractions 44.8% Late liponecrosis 14.8%8.3 (Patient reported) 8.4 (Physician reported) (Assessed by VAS with 0–10 score)	-

Abbreviations: LRR = local recurrence rate; IBTR = in-breast tumor recurrence; CSS = cancer specific survival; RFS = relapse-free survival; OS = overall survival; DSS = disease-specific survival; HDR BCT = high dose rate brachytherapy; PDR BCT = pulse dose rate brachytherapy; LRFS = local relapse free survival; DFS = disease-free survival; LDR BCT = low dose rate brachytherapy; IBR = in-breast recurrence; BC = breast cancer; MFS = metastases free survival; LR = local recurrence; LC = local control; IBF = in-breast failure. Cosmesis results are recorded according to the Harward breast cosmesis scale.

**Table 4 cancers-14-02564-t004:** Main studies concerning partial breast re-irradiation with brachytherapy.

Study	Type of Study	Number of Patients	Follow Up (m)	Total Dose (Gy) (Dose for Fraction) and Technique	Outcomes: OS, 3rtIBTE-FS §	Toxicities > G3 (%)	Cosmesis Result
Maulard C. et al. (1995) [80].	Retrospective	38	48	30 MIB-LDR	5y-OS: 55% § 21%	8 (2 pts: skin necrosis, 1 pts: severe breast pain)	4 pts: good, 20 pts: acceptable, 9 pts: mediocre.
Hannoun-Levi J.M. et al. (2004) [81].	Retrospective	69	50.2	30–50 MIB-LDR	91.8% (5y-OS) § 77.4%	10.2 (2 Pts: necrosis requiring surgery)	Not reported
Niehoff P. et al. (2006) [82].	Retrospective	19	36	28 (2.5 BID) MIB-HDR/PDR	68.7% (1y-OS) § 62.5%	5 (1 pts: skin ulceration)	Not evaluated
Chadha M. et al. (2008) [83].	Retrospective	15	36	30–45 MIB-LDR	100% (3y-OS) § 89%	0	100% pts: Good or excellent
Guix B. et al. (2009) [84].	Retrospective	36	89	30 (2.5 BID)	96.7% (10y-OS) § 89.4%	0	96% pts: satisfactory
Hannoun-Levi J.M. et al. (2011) [85].	Retrospective	42	21	34 (3.4 BID) MIB-HDR	not reported § 3%	Not reported	100% pts: satisfactory
Kauer-dorner D. et al. (2012) [86].	Prospective	39	57	34 (0.6–1) MIB-PDR	87% (5y-OS) § 93	16.7 (1pts: breast fibrosis. 32pts: pain)	3pts: excellent, 6 pts: good, 9pts: fair, 1pts: unacceptable
Hannoun-Levi J.M. et al. GEC-ESTRO (2013) [11].	Prospective	217	47	32 (4 BID) MIB-HDR 50 (0.6–1) MIB-LDR	76.4% (10y-OS) § 94.4%	10% grade 31% grade 4 (ulceration)	52 pts: excellent, 40 pts: good, 14 pts: fair, 2 pts: poor
Smanyko V. et al. (2019) [87].	Prospective	195	52	22 (4.4 BID) MIB-HDR	81% (5y-OS) § 94%	8	70%: good
Montagne L. et al. (2019) [88].	Retrospective	159	71	28–34 MIB-HDR 30–55 MIB-LDR	91.2% (6y-OS) § 97.4%	1 pts: grade 4 ulceration	122 pts: excellent/good. 6 pts: poor
Forster T. et al. (2019) [89].	Retrospective	19	65	49.8–50.4 (0.5–0.7) MIB-LDR 34.2–32 (3.4–3.8) MIB-HDR	100% (10y-OS) § 100%	0	Not reported
Cozzi S. et al. (2019) [90].	Retrospective	40	61.5	32–34 (4–3.4 BID) MIB-HDR	85.3% (5y-OS) § 96%	3pts: 11 pts fibrosis G3, 3pts: fibrosis G4	100% satisfactory
Vavassori A. et al. (2020) [91].	Retrospective	31	73.7	34 (3.4 BID) MIB-HDR	87.1% 85y-OS § 90.3%	0	100% good
Chatzikonstantinou G. et al. (2021) [92].	Retrospective	20	69.6	32 (4 BID) MIB-HDR	92.3% (5y-OS) § 86.8%	0	6pts: excellent, 6pts: good, 3 pts: fair, 1 pt: poor

Abbreviations: Gy: Gray, pts: patients, OS: overall survival, 3rdIBTE: third ipsilateral breast tumor event rate, G3 tox.: grade 3 and higher toxicity rate, MIB: Multicatheter interstitial brachytherapy, LDR: low-dose rate, PDR: pulsed-dose rate, HDR: High-dose rate, y: years. Cosmesis results are recorded according to the Harward breast cosmesis scale.

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
