# Peer review of "The Role of Interstitial Brachytherapy for Breast Cancer Treatment: An Overview of Indications, Applications, and Technical Notes"

_cancers, 2022, doi:10.3390/cancers14102564_

Round 1

Reviewer 1 Report

Dear Authors,

I congratulate you on the interesting topic. You provide a well-written overview of BCT by describing its indications, applications, advantages and limits, also providing some notes about the technique itself. The paper is interesting and provide a useful background for further in-depth evaluations and clinical trials about the use of this radiation technique.

Given these considerations, the contribution of your paper, at this stage, is sufficiently developed such that it could be accepted for publication in Cancers after a minor spell check and correction of typos (f.e. lines 28, 29, toxicities instead of toxixities in tables, excellent instead of exellent in tables, etc.).

Furthermore, in table 2, 3 and 4, in the section “Cosmesis Results”, please, explain if you have added the results or a legend for the results provided in a different section.

Finally, in table 4, what <*> refers to?

Kind Regards

Author Response

I congratulate you on the interesting topic. You provide a well-written overview of BCT by describing its indications, applications, advantages and limits, also providing some notes about the technique itself. The paper is interesting and provide a useful background for further in-depth evaluations and clinical trials about the use of this radiation technique.

Given these considerations, the contribution of your paper, at this stage, is sufficiently developed such that it could be accepted for publication in Cancers after a minor spell check and correction of typos (f.e. lines 28, 29, toxicities instead of toxixities in tables, excellent instead of exellent in tables, etc.).

Thank you for the comments.

We apologize for the typos, we have read again the article and corrected the errors. Thanks

Furthermore, in table 2, 3 and 4, in the section “Cosmesis Results”, please, explain if you have added the results or a legend for the results provided in a different section.

We added in the caption of the tables that the evaluation of the cosmesis results are recorded according to the Harward breast cosmesis scale

Finally, in table 4, what <*> refers to?

sorry, it was an error, we have removed the <*> symbol

Reviewer 2 Report

The authors have compiled a very constructive and informative review entitled"The role of interstitial brachytherapy for breast cancer treatment:an overview of indications,applications and technical notes".The impantation technique and treatment delivery is described in detail alongwith the brachytherapy doses.The adavantages and disadvantages have been dealt with in quite detail.

The tables presented in the review are very informative and portray useful data.

This review is clinically very relevant and as well as for breast cancer patients and researchers.

Author Response

The authors have compiled a very constructive and informative review entitled"The role of interstitial brachytherapy for breast cancer treatment:an overview of indications,applications and technical notes".The impantation technique and treatment delivery is described in detail alongwith the brachytherapy doses.The adavantages and disadvantages have been dealt with in quite detail.

The tables presented in the review are very informative and portray useful data.

This review is clinically very relevant and as well as for breast cancer patients and researchers.

Thanks for the comments.

Reviewer 3 Report

I find this an interesting and well written review on breast brachytherapy.

However, I would suggest the authors to more clearly present the main indications and contraindications for the breast brachytherapy. When is it preferred over whole breast irradiation, and when is whole breast irradiation the better option? This could be either emphasised in the discussion or lined out in the conclusion part.

Other than minor spell-checking, I have no further objections and find this review worth of publishing.

Author Response

I find this an interesting and well written review on breast brachytherapy.

However, I would suggest the authors to more clearly present the main indications and contraindications for the breast brachytherapy. When is it preferred over whole breast irradiation, and when is whole breast irradiation the better option? This could be either emphasised in the discussion or lined out in the conclusion part.

Other than minor spell-checking, I have no further objections and find this review worth of publishing.

Thanks for the comments.

We add in line 329: “It should be emphasized that in cases of tumors larger than 3 cm, or the presence of axillary or internal mammary chain node metastases, or in the case of breast implants or expanders, brachytherapy should not be performed and whole breast external beam radiotherapy is mandatory”.

WE also provided a correction of typos.